# Urinary and Daily Assumption of Polyphenols and Hip-Fracture Risk: Results from the InCHIANTI Study

**DOI:** 10.3390/nu14224754

**Published:** 2022-11-10

**Authors:** Raffaello Pellegrino, Roberto Paganelli, Stefania Bandinelli, Antonio Cherubini, Cristina Andrés-Lacueva, Angelo Di Iorio, Eleonora Sparvieri, Raul Zamora-Ros, Luigi Ferrucci

**Affiliations:** 1Department of Scientific Research, Campus Ludes, Off-Campus Semmelweis University, 6912 Lugano, Switzerland; 2Internal Medicine, UniCamillus, International Medical University, 00133 Rome, Italy; 3Geriatric Unit, Azienda Toscana Centro, 50125 Florence, Italy; 4Geriatrics and Geriatric Emergency Care, Italian National Research Center on Aging (IRCCS-INRCA), 60126 Ancona, Italy; 5Biomarkers and Nutrimetabolomics Laboratory, Nutrition, Food Science and Gastronomy Department, Faculty of Pharmacy and Food Science, University of Barcelona, 08035 Barcelona, Spain; 6Department of Innovative Technologies in Medicine & Dentistry, University “Gabriele d’Annunzio”, 66100 Chieti, Italy; 7Internal Medicine Unit, Ospedale S. Liberatore, 64100 Teramo, Italy; 8Unit of Nutrition and Cancer, Epidemiology Research Program, Catalan Institute of Oncology, Bellvitge Biomedical Research Institute (IDIBELL), 08035 Barcelona, Spain; 9Longitudinal Studies Section, Translational Gerontology Branch, National Institute on Aging, National Institutes of Health, Baltimore, MD 21224, USA

**Keywords:** bone-pQCT, dietary polyphenols, longitudinal study, urinary polyphenols, femur fracture risk

## Abstract

A high polyphenol intake has been associated with higher bone-mineral density. In contrast, we recently demonstrated that the urinary levels of these micronutrients were associated with the long-term accelerated deterioration of the bone. To expand on the health consequences of these findings, we assessed the association between urinary level and dietary intake of polyphenols and the 9-year risk of hip fractures in the InCHIANTI study cohort. The InCHIANTI study enrolled representative samples from two towns in Tuscany, Italy. Baseline data were collected in 1998 and at follow-up visits in 2001, 2004, and 2007. Of the 1453 participants enrolled at baseline, we included 817 participants in this study who were 65 years or older at baseline, donated a 24 hour urine sample, and underwent a quantitative computerized tomography (pQCT) of the tibia. Fracture events were ascertained by self-report over 9 years of follow-up. Thirty-six hip fractures were reported over the 9-year follow-up. The participants who developed a hip fracture were slightly older, more frequently women, had a higher dietary intake of polyphenols, had higher 24-hour urinary polyphenols excretion, and had a lower fat area, muscle density, and cortical volumetric Bone Mineral Density (vBMD) in the pQCT of the tibia. In logistic regression analyses, the baseline urinary excretion of total polyphenols, expressed in mg as a gallic acid equivalent, was associated with a higher risk of developing a hip fracture. Dietary intake of polyphenols was not associated with a differential risk of fracture. In light of our findings, the recommendation of an increase in dietary polyphenols for osteoporosis prevention should be considered with caution.

## 1. Introduction

The aging of the population leads to the increased incidence and prevalence of chronic morbidity, which imposes high costs of care and challenges the quality of life of patients and their caregivers. The identification of risk factors for chronic diseases that can be prevented through campaigns that promote positive health behaviors play a crucial role in managing this trend. A gradual loss of bone density and quality occurs with aging and results in progressive osteopenia and osteoporosis [1,2]. Osteoporosis substantially increases the risk of skeletal fractures and further morbidity and mortality [3]. The diagnostic criteria for osteoporosis include a DEXA bone-mineral density (BMD) measurement ≥2.5 (SD) below the average value for young healthy women (T-score ≤ −2.5 SD), while osteopenia is characterized by a borderline decrease in BMD and is defined by a BMD T-score between −1.0 and −2.5 [4]. Preventive measures, apart from pharmacologic agents, include lifestyle adjustments, nutritional support, fall prevention strategies, and exercise [5]. Most clinical practice guidelines suggest that diets rich in calcium and vitamin D may limit osteoporosis progression [6,7], and when dietary sources are insufficient or poorly tolerated, pharmacological supplementation could be useful. However, this recommendation has been challenged in recent systematic reviews and meta-analyses [8,9]. Another lipophilic vitamin, menaquinone-7 (vitamin-K2), has been also suggested as potentially useful in preventing osteoporosis [10]; however, its high cost and the lack of agreement on optimal dosages have somewhat limited its widespread use.

Polyphenols, bioactive compounds derived from plant-based foods, have antioxidant and anti-inflammatory properties. It has been proposed that an increased intake of polyphenols can effectively slow down the osteoporosis process because of their bone anabolic action [11,12]. Among those polyphenols, naringenin, kaempferol, luteolin, quercetin, epigallocatechin 3-gallate, and resveratrol were reported to be effective in bone metabolism, at least in animal model studies [13,14]. Epidemiological studies have shown that at least five cups/day of tea, rich in polyphenols as much as in caffeine, are positively associated with higher bone-mineral density (BMD) at multiple skeletal sites, while the relationship with fracture risk is less clear [15,16]. Studies in humans have shown either a reduction or no difference in the risk of fragility fracture with tea consumption [15,17].

Contrary to this view, in the large population-based InCHIANTI study, we recently reported that higher urinary polyphenols were associated with a long-term accelerated deterioration of bone health (bone mass, diaphyseal design, and material quality), measured by peripheral quantitative computed tomography (pQCT) [18]. At least two hypotheses must be considered to explain those results: polyphenols form robust, even reversible complexes with ions, and in particular with iron and calcium (a higher consumption of polyphenols is associated with a lower prevalence of kidney stones); and the decline in glomerular filtration rate (GFR) with aging that could reduce the bioavailability of serum calcium [18]. However, the described deterioration of bone mass, diaphyseal design, and material quality associated with a higher polyphenol intake does not necessarily imply an increase in bone fracture. Therefore, the aim of this study was to evaluate the association of incident femur fractures with urinary total polyphenols (UTPs) and total polyphenol intake/day, in a cohort of free-living subjects, representative of the over-65 Italian population.

## 2. Materials and Methods

The design of the InCHIANTI study has been described in detail elsewhere [19]. Briefly, the study was designed by the Laboratory of Clinical Epidemiology of the Italian National Institute of Research and Care on Aging (INRCA, Florence, Italy), and was performed in two small towns in Tuscany. The baseline data were collected in 1998–2000, the three-year follow-up took place in 2001–2003, the six-year follow-up took place in 2004–2006, and the nine-year follow-up took place in 2007–2009.

### 2.1. Samples

Of the 1453 participants enrolled at baseline in the InCHIANTI study, 817 subjects were included in this study because they were 65 years or older, had at least one follow-up, and all variables of interest for this study were available. Participants were all European subjects and of Caucasian race. The Ethical Committee of the Local Health Authority of Florence, Tuscany Region, approved the study protocol, and written informed consent was obtained from each participant. From the InCHIANTI study population, we selected all participants who reported an incident hip fracture during the follow-up. The subjects that reported a hip fracture at baseline were excluded from the analysis. Moreover, the few cases who experienced a bilateral or repeated hip fracture were counted only once.

### 2.2. Dietary Assessment

At baseline, as in the subsequent follow-ups, usual food consumption and energy intake were estimated through personal interviews and by the Italian version of the European Prospective Study into Cancer and Nutrition (EPIC) [20,21]. For the purpose of this study, the following variables were used: the number of glasses of wine consumed daily; and the intake of total dietary polyphenols (TDPs) consumed (mg/day), which was calculated according to Zamora-Ros as the sum of flavonoids, phenolic acids, lignans, stilbenes, and other polyphenols expressed as aglycone equivalents (mg/day) [22].

### 2.3. Urinary Total Polyphenols (UTPs)

At baseline, 24 hour urine samples were obtained from the participants. Urine samples were aliquoted and stored at −80 °C until analysis. Samples were thawed on ice and analyzed using the Folin–Ciocalteau (F–C) assay after solid-phase extraction, which yields the elimination of interfering substances that could react with the F–C assay, as described previously [23]. Intra-batch and inter-batch coefficients of variation were less than 10.5% and less than 10.7%, respectively [24]. UTP concentrations were expressed as mg of gallic acid equivalents (GAE) per 24-hour versionurine sample.

### 2.4. Tibial pQCT

Peripheral quantitative computed tomography (pQCT) was performed by the XCT 2000 device (Stratec Medizintechnik, Pforzheim, Germany). The description of the pQCT examination in the InCHIANTI study has been published elsewhere [1]. The images obtained from the pQCT were analyzed using BonAlyse software version 1.2 (BonAlyse Oy, Jyvaskyla, Finland). The following bone parameters were derived [25]: cortical bone area (CBA): area of the cortical bone region of the tibia cross-section in cm^2^; total volumetric mineral density (vBMDtot) assessed as the average density of the total bone area in mg/cm^3^; cortical volumetric mineral density (vBMDc), a selective measure of the apparent volumetric density of cortical bone, in mg/cm^3^; cortical thickness (CTh), the average thickness of the circular crown formed by the centered periosteal and endocortical circumferences, in mm; medullary area (MedA), which is the difference between total and cortical bone areas and includes the marrow space and areas of the inner cortex trabecularized by endocortical resorption that has a cortical apparent vBMD < 710 mg/cm^3^ and is sensitive to endocortical resorption, in cm^2^; tibia cross-section total bone area in cm^2^. From the pQCT images measured at 4% tibia length, the following bone parameters were obtained: bone circumference (mm); total bone area (mm^2^); and cortical vBMD (mg/cm^3^). Using the pQCT images obtained at 66% tibia length from the distal tip of the tibia, we also estimated the calf muscle cross-sectional area (CMCSA; expressed in cm^2^), muscle density, and the bone–muscle ratio, both expressed in mg/cm^3^. Lastly, from a scan performed at 90% tibia length, the maximal fat area was calculated and expressed in mm^2^.

### 2.5. Covariates

#### 2.5.1. Laboratory Tests

Blood samples were collected in the morning after a 12-hour fast, centrifuged, and stored at −80 °C. Serum creatinine levels (mg/dL) were measured by the Laboratory of Clinical Chemistry and Microbiological Assays, SS., Annunziata Hospital, Azienda Sanitaria 10, Florence, Italy, using a colorimetric assay (TP, Roche Diagnostics, GmbH, Mannheim, Germany) and a Roche analyzer (Roche Diagnostics, GmbH, Mannheim, Germany). At baseline, the analyzer used was a Hitachi 917. For the follow-ups, it was a Modular P800 Hitachi. The glomerular filtration rate was calculated according to the Cockcroft–Gault formula [26].

#### 2.5.2. Physical Performance and Strength

The short physical performance battery (SPPB), based on lower-extremity performance tests, was used to summarize lower-extremity performance [27]. The SPPB consisted of walking speed, ability to stand from a chair, and ability to maintain balance in progressively more challenging positions. Each physical performance measure was categorized into a five-level score, with 0 representing the inability to conduct the test and 4 representing the highest level of performance. The score registered in the three measures were then added to create a summary measure ranging from 0 (worst) to 12 (best).

Handgrip strength was measured using a handheld dynamometer (hydraulic hand “BASELINE”; Smith & Nephew, Agrate Brianza, Milan, Italy). Participants were asked to perform the task twice with each hand, and the results were averaged.

Information on smoking status was collected at home interviews. BMI (in kg/m^2^) was calculated using measured weight (in kg) divided by height (in m^2^), both of which were measured during the medical visit.

### 2.6. Statistical Analysis

The urinary level of gallic acid and the intake of total dietary polyphenols were not normally distributed and were therefore normalized. The urinary level of gallic acid was cubic-root normalized (Skewness *p*-value = 0.84 and Kurtosis *p*-value = 0.15), whereas total dietary polyphenols were natural-logarithm transformed (Skewness *p*-value = 0.10 and Kurtosis *p*-value = 0.32). The baseline characteristics were compared between groups for all the variables of interest, using analysis of variance for the continuous variables and χ^2^ test analyses for the dichotomous or categorical variables; moreover, in the descriptive table, we reported the *p*-values adjusted for age and sex using linear and logistic regression models, respectively. In the descriptive analysis, the normalized gallic acid urinary eq/day *p*-value was also adjusted for 24-hour creatinine clearance. The logistic regression model was used to assess the risk of developing hip fractures associated with the baseline urinary level of gallic acid considered as a continuous variable, adjusted for potential confounders that were associated with the risk of fracture at univariate analysis with a *p*-value < 0.10. Creatine clearance at 24-hour was forced into the model to counteract its confounding effect on the gallic acid urinary level. The lower median values for muscle density at 66% of tibia length (mg/cm^3^), volumetric bone density BDG at 4% of tibia length (mg/cm^3^), and fat area at 90% of tibia length (mm^2^) were considered in the logistic model as the reference group. Due to not normally distribution of naringenin, kaempferol, luteolin, quercetin, epigallocatechin 3-gallate, and total resveratrol, and due to the difficulty in their normalization, difference comparisons between the groups (hip fracture, sex, and alcohol consumption) were evaluated through quantile regression to model the effects of covariates on the conditional quantiles of a response variable. To assess which factors were associated with TDP variation across the time of the study, a linear mixed model was analyzed, in which UTPs, age, male sex compared to female, and history of hip fracture were independent variables. Analyses were conducted using SAS 9.4 (SAS Institute Inc., Cary, NC, USA).

## 3. Results

Thirty-six hip fractures were detected over a 9-year follow-up (4.4%); subjects who had a femur fracture were slightly older, more likely to be women (*p* = 0.01), reported a higher dietary intake of polyphenols (*p* = 0.01), and had higher values of 24-hour urinary excretion of gallic acid, independent of sex or age (*p* = 0.05) (Table 1).

During the pQCT, participants who developed a hip fracture, compared to those who did not, had a smaller fat area (90% of tibia length), lower muscle density (66% of tibia length) and lower cortical vBMD (4% of tibia length) (Table 2).

The development of a hip fracture was not associated with the prevalence and incidence of other morbidities, nor in the number of prescribed drugs, including those relevant for osteoporosis (vitamin D, bisphosphonates, or teriparatide (Appendix A)).

Table 3 shows the results of the logistic regression analysis: in Model 1, we evaluated the role of the urinary concentration of gallic acid in predicting the risk of hip fractures, while in Model 2, the effect of total polyphenols consumed daily was evaluated. Urinary gallic acid values were associated with higher fracture risks OR = 2.10; 95% CI: 1.16–3.81. The effect of the polyphenols was independent from the effect of other potential confounders included in the models, such as age, sex, BMD at 4% of tibia length, the area of adipose tissue at 90% of tibia length, and muscle density at 66% of tibia length. A greater muscle density was associated with a lower fracture risk [OR = 0.91; 95% CI: 0.83–0.99]. On the contrary, the daily dietary intake of polyphenols estimated by the EPIC questionnaire was not associated with hip-fracture risk.

To assess beyond the total polyphenol daily intake, the possible association of specific polyphenol intake levels with hip-fracture risk and differences in the median and 95%CI were evaluated for naringenin, kaempferol, luteolin, quercetin, epigallocatechin 3-gallate, and total resveratrol, which were age- and sex-adjusted (Table 4). Only the daily intake resveratrol showed higher and statistically significant levels in subjects without a hip fracture compared to those with fractures [−0.08 (−0.1:−0.0); *p*-value = 0.005]. Looking at a gender effect, we conducted a stratified analysis for sex. Male subjects showed a statistically significant higher level of daily intake of resveratrol compared to female subjects [0.18 (0.11:0.21); *p*-value < 0.001] age-adjusted. Lastly, to assess the potential confounding effect of alcohol consumption between sex differences, we also stratified this voluptuary. After adjusting for age and sex, teetotal subjects showed statistically significant lower levels of resveratrol, compared to the prior or current alcohol users group [−0.12 (−0.15:−0.09) *p*-value < 0.001]. A multiplicative gender for alcohol effect could not be assessed, probably due to the limited male sample that experienced hip fractures. Lastly, to verify role of UTPs independent of resveratrol intake and alcohol consumption in predicting the risk of hip fractures, we introduced resveratrol and alcohol intake as covariates in the logistic model predicting hip fractures, but the strength of the predictive risk of hip fractures associated with UTPs was practically unchanged [OR = 2.11; 95%CI (1.15–3.89); *p*-value = 0.01].

To assess factors associated with TDP variation through the times of the study, linear mixed model analyses were conducted (Table 5). TDPs consumed did not vary across the time of the study (*p*-value = 0.39); UTPs were strongly associated with TDPs (*p*-value = 0.004), and no multiplicative effect could be demonstrated for the interaction with time (*p* = 0.77); with increasing age, lower UTP levels were found (*p*-value = 0.002). Lastly, the male sex showed higher levels of TDPs compared to the female sex (*p*-value < 0.001).

## 4. Discussion

The main result of this study was that high urinary polyphenol levels in the InCHIANTI population aged 65 and over were associated with a higher risk of developing hip fractures. The risk appears to be almost double for subjects with higher UTPs compared to those with lower values. Moreover, consistent with the previous literature, greater muscle density, higher median vBMD, and larger median fat area, represented independent protective factors against the development of hip fractures.

To the best of our knowledge this is the first prospective study of a large cohort of free-living subjects that assesses the role of urinary polyphenols in the risk of incident hip fractures. Several previous works have examined the potential effect of polyphenols on bone quality [28]. Epidemiological studies on tea consumption (as a source of polyphenols) demonstrated that tea could be a promising approach for increasing BMD [16]. Additionally, diet habits, namely the Mediterranean diet [29] and a regular consumption of fruit [30], have been reported to positively affect bone quality [26]. Recently, findings from the InCHIANTI study provided evidence that higher UTP levels were associated with the long-term accelerated deterioration of bone health [18]. In order to assess the level of bone health, several bone markers can be used, such as bone mass, material quality, and diaphyseal design; however, a low bone quality does not necessarily imply a higher risk of fracture, but subjects with low bone quality (osteopenia) suffer more fractures compared to osteoporotic subjects [31]. Several factors may contribute to this apparent paradox; among them is the low specificity of BMD as a predictor of femur and vertebral frailty fractures. Therefore, a higher BMD may not correlate with a reduced incidence of fracture. The evidence for the role of polyphenols in modulating the risk of fracture is at best not conclusive. Most research has focused on experimental animal models, demonstrating that polyphenols increase bone quality and could reduce the time of recovery from a fracture [32]. Clinical trials involving the oral supplementation of polyphenols in human subjects showed an enhancement of bone quality but failed to provide solid proof of a reduction in incident fracture risk [33,34]. On the contrary, our study clearly demonstrates an increase in fracture risk for the urinary polyphenols analyzed, independent from all the confounders considered. However, the total polyphenol intake in the diet did not correlate with the risk of incident fractures. Only resveratrol among the many polyphenols assumed daily and considered in the analysis showed higher levels in those subjects with incident hip factures. Additionally, in those subjects who reported to drink alcohol, resveratrol could not modulate the strength of the association between the urinary level of gallic acid and fracture risk. All together, these results indicate that resveratrol assumed daily is an indirect marker of alcohol consumption, which is a well-known factor risk of osteoporosis and hip fractures. It is difficult to speculate on the mechanism(s) which may account for our results. We may consider at least two hypotheses: the high affinity of polyphenols for iron and calcium [35]; and the decline of the glomerular filtration rate with aging [36], which might interfere with bone homeostasis and could facilitate the urinary formation of polyphenol–calcium complexes, thus reducing the bioavailability of calcium [36]. This scenario is not in agreement with a recent meta-analysis showing a reduction in serum calcium levels after resveratrol supplementation [37].

The still incomplete understanding of polyphenol metabolism is most probably the reason for these apparently conflicting results. The role of the gut microbiome on the metabolic transformation and absorption of polyphenols has only recently been recognized [38]. Therefore, the divergent roles of TDPs and UTPs in fracture risk could also be due to a phenotypical predisposition to an increased urinary excretion of polyphenols, which may translate into an increased risk of osteoporosis. This may represent an important limitation of our study, i.e., the use of UTP values as markers of polyphenol exposure, which may not be consistent with blood levels and/or dietary intake. UTPs were only measured at baseline, and this single measurement might be insufficient to fully assess the role of polyphenols in predicting bone health and fracture risk. Moreover, subjects’ dietetic style could vary during life, and consequently polyphenol assumption could also vary. Therefore, the hip-fracture risk assessed in our analysis could be somehow overestimated. However, in our results, as in previous work [22], demonstrated a strong correlation between UTPs and TDPs, also independently from time (Table 5) and we could consider the UTP level constant during the time of the study. However, these limitations can be dismissed because the urinary assessment of polyphenols is a reliable and validated method [22]. An additional limitation of our study is the lack of consideration for fractures other than the femur, and the risk might be underestimated in this case. Finally, in our study, bone turnover markers were not considered, which have emerged as promising tools in the management of osteoporosis independently from bone-mineral density (BMD) measurements [39].

## 5. Conclusions

This study demonstrated that higher urinary polyphenol levels are predictive of a greater risk of incident fractures in a representative, free-living Italian population. More extensive information is needed on the metabolism and effects of polyphenols on bone homeostasis and confirmed by ad hoc intervention clinical trials. In the light of the present results, the recommendation of dietary polyphenol supplementation for osteoporosis prevention should be considered with caution.

## Figures and Tables

**Table 1 nutrients-14-04754-t001:** Baseline characteristics of the population according to incident hip fractures. All *p*-values were age- and sex-adjusted. Normalized gallic acid urinary eq/day *p*-value was also adjusted for 24 hour creatinine clearance.

	Hip Fracture		*p*-Value
	No	Yes	
	781	36	
Age (years)	75.38 ± 7.64	76.75 ± 7.35	0.20
Sex female (*n*, %)	426 (54.6)	27 (75.0)	0.01
Alcohol consumption			0.92
Teetotal	123 (15.6)	6 (16.7)	
Former	175 (22.4)	7 (19.4)	
Current	483 (61.8)	23 (63.9)	
Cigarette smoking			0.61
Never	462 (59.2)	22 (61.1)	
Former	210 (26.9)	11 (30.6)	
Current	109 (13.9)	3 (8.3)	
Summary Performance Score (0–12)	9.77 ± 3.28	8.70 ± 3.39	0.20
Body Mass Index (kg/m^2^)	27.49 ± 4.07	26.74 ± 4.77	0.22
Muscle strength (kg)	28.74 ± 12.04	25.17 ± 9.48	0.71
Normalized gallic acid urinary eq/day ^1^	5.28 ± 0.74	5.43 ± 0.70	0.05
Normalized Tot. polyphenols consumed (mg/day) ^2^	443.84 ± 165.88	467.17 ± 166.55	0.01
Creatinine clearance, 24-hour (mL/min)	76.00 ± 25.71	69.99 ± 27.69	0.73

^1^ Gallic acid was cubic-root normalized. ^2^ Total polyphenols consumed was natural-logarithm transformed.

**Table 2 nutrients-14-04754-t002:** Baseline pQCT results according to incident femur fractures. All *p*-values were age- and sex-adjusted.

	Hip Fracture		*p*-Value
	No	Yes	
	781	36	
Fat area 90% tibia (mm^2^)	2631.46 ± 1521.81	2451.89 ± 1381.59	0.007
Muscle area 66% tibia (mm^2^)	6233.66 ± 1255.49	5865.70 ± 1417.17	0.79
Muscle density 66% tibia (mg/cm^3^)	70.59 ± 3.66	69.08 ± 5.88	0.05
Cortical bone area 38% tibia (mm^2^)	295.73 ± 75.31	275.33 ± 78.27	0.79
Volumetric BMD 38% tibia (mg/cm^3^)	474.88 ± 45.49	479.21 ± 41.78	0.40
Volumetric Cortical BMD 38% tibia (mg/cm^3^)	1097.97 ± 78.68	1080.51 ± 81.67	0.52
Total bone area 38% tibia (mm^2^)	380.09 ± 71.51	361.26 ± 76.18	0.94
Medullar area 38% tibia (mm^2^)	86.06 ± 32.13	85.93 ± 29.03	0.61
Cortical bone thickness 38% tibia (mm)	4.62 ± 1.24	4.34 ± 1.70	0.67
Bone/muscle ratio 66% tibia	0.10 ± 0.03	0.11 ± 0.04	0.57
Total vBMD 4% tibia (mm^2^)	1079.87 ± 382.20	1041.26 ± 425.54	0.37
Cortical vBMD 4% tibia (mg/cm^3^)	256.93 ± 52.96	228.33 ± 38.39	0.01
Bone circumference BDG 4% tibia (mm)	135.59 ± 37.51	139.59 ± 38.87	0.21

**Table 3 nutrients-14-04754-t003:** Logistic regression analysis, factors predicting fracture risk. Urinary gallic acid was cubic-root normalized; total polyphenols consumed was natural-logarithm transformed. Fat area at 90% tibia length, volumetric bone density BDG at 4% tibia length, and muscle density at 66% tibia length were contrasted in the analysis according to median level of distribution.

	Model 1		Model 2	
	OR (95%CI)		OR (95%CI)	
Normalized Gallic acid eq/day ^1^	2.06 (1.12–3.76)	0.01		
Total polyphenols consumed daily (mg/day) ^1^			1.71 (0.49–5.96)	n.s.
Sex				
Female	5.04 (1.53–16.6)	0.008	4.92 (1.43–16.89)	0.02
Male	reference group		reference group	
Age (years)	0.99 (0.94–1.06)	n.s.	0.99 (0.93–1.06)	n.s.
Fat area 90% tibia (mm^2^)				
Upper median	0.32 (0.13–0.77)	0.01	0.36 (0.15–0.85)	0.02
Lower median	reference group		reference group	
Volumetric bone density BDG 4% tibia (mg/cm^3^)				
Upper median	0.05 (0.01–0.46)	0.009	0.06 (0.01–0.54)	0.01
Lower median	reference group		reference group	
Muscle density 66% tibia (mg/cm^3^)				
Upper median	0.01 (0.01–0.64)	0.03	0.01 (0.01–0.76)	0.04
Lower median	reference group		reference group	
Creatinine Clearance 24-hour ^1^	0.99 (0.98–1.02)	n.s.	1.01 (0.98–1.02)	n.s.

^1^ risk of femur fracture for one-unit increase. N.S. *p*-value > 0.05

**Table 4 nutrients-14-04754-t004:** Quantile regression models. Polyphenols consumed daily at baseline according to follow-up incident hip fracture. Data were reported as differences of the median estimates and 95%CI. All *p*-values were age- and sex-adjusted. The reference group was represented by subjects without hip fracture.

	Estimate (95%CI)	*p*-Value
Naringenin (mg/day)	−2.18 (−5.7:1.4)	0.23
Kaempferol (mg/day)	0.28 (−0.1:0.60)	0.09
Luteolin (mg/day)	0.02 (−0.3:0.3)	0.91
Quercetin (mg/day)	−0.10 (−0.2:0.1)	0.09
Epigallocatechin 3-gallate (mg/day)	−0.28 (−2.6:2.1)	0.81
Total resveratrol (mg/day)	−0.08 (−0.1:−0.0)	0.005

**Table 5 nutrients-14-04754-t005:** Linear mixed model. Factors associated with total dietary polyphenols (TDPs) assumption, variation through the times of the study assessed with European Prospective Study into Cancer and Nutrition s(EPIC) questionnaire. Urinary total polyphenol (UTP) concentrations were expressed as normalized mg of gallic acid equivalents (GAE) per 24-hour urine. Linear Mixed Model estimates are reported as β ± SE.

	β ± SE	*p*-Value
Intercept	592.49 ± 80.53	<0.001
Time	−18.24 ± 21.13	0.39
UTP normalized gallic acid	24.53 ± 8.47	0.004
Time for UTP normalized gallic acid	−1.16 ± 3.92	0.77
No hip fracture	−17.22 ± 21.82	0.44
Age (years)	−2.27 ± 0.73	0.002
Sex (male)	57.09 ± 9.35	<0.001

## Data Availability

The datasets used and/or analyzed during the current study are available from the responsible authors for the InCHIANTI study (Luigi Ferrucci) on reasonable request. Data of the InCHIANTI study is available to all researchers upon justified request using the proposal form available on the InChianti website (http://inchiantistudy.net/wp/how-to-submit-a-proposal/, accessed on 8 October 2022).

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
