# Peer review of "Urinary and Daily Assumption of Polyphenols and Hip-Fracture Risk: Results from the InCHIANTI Study"

_nutrients, 2022, doi:10.3390/nu14224754_

Round 1

Reviewer 1 Report

1. I suggest adding osteoporosis and osteopenia criteria in the introduction. 

2. Why do you use capital letter for name of polyphenols ("among those polyphenols, Naringenin, Kaempferol, Luteolin, Quer- 63 cetin, Epigallocatechin 3-gallate, and Resveratrol")

3. You assessed polyphenols intake at baseline and follow-up. However you do not know how many polyphenols subjects intake during 8 years. Maybe, somebody has changed habits after one year or 8 year, which may affects your results. 

4. Introduction is really poor. Maybe good point is adding information about content of polyphenols in Medditerranean diet. 

Author Response

First of all, we have to thanks this reviwer suggestions and efforts to ameliorate our manuscript.

  1. I suggest adding osteoporosis and osteopenia criteria in the introduction (line 49). 

Accordingly with this reviewer suggestion we insert the definition of osteopenia/osteoporosis t-score, according to WHO.

  1. Why do you use capital letter for name of polyphenols ("among those polyphenols, Naringenin, Kaempferol, Luteolin, Quercetin, Epigallocatechin 3-gallate, and Resveratrol")

We apology, and accordingly to this reviewer suggestion we amended the text. 

  1. You assessed polyphenols intake at baseline and follow-up. However you do not know how many polyphenols subjects intake during 8 years. Maybe, somebody has changed habits after one year or 8 year, which may affects your results. 

Both reviewers # 1 e #3 arise a similar observation, we could not know what happened to polyphenols assumption during the 9 years follow-up, as to other confounders.

This is only partially true; we have repeated measures for almost all the variables considered in the manuscript. In our recent paper we have found a decrease in the bone quality related to high UTP levels, but BMD is not enough to predict bone fractures. Therefore, we want to verify if high level of UTP could be a marker of hip fracture increased risk. The problem is similar, for example, to the evaluation of the heart attack risk and smoking status, if the patient stops smoking obviously the heart attack risk decrease; even though smoke habit remained a risk factor.

Unfortunately, we have only basal UTP assessment, whereas for daily P assumption we have a measurement in every follow-up. Therefore, we conducted a new analysis that we reported at the end of the result section, to demonstrate that:

  1. UTP well correlate with P. daily assumption through study times;
  2. P did not change statistically significant through time of the study;
  3. And lastly as we could expect, accordingly to our results, daily dietary-assumption of P, did not differ between patients reporting or not hip fracture.

We did not consider any other confounders variation since in the logistic final model they were not analyzed, because they were not statistically significant different between the two study groups at baseline (David W. Hosmer, Stanley Lemeshow; Applied Logistic Regression. Willey Ed.)

  1. Introduction is really poor. Maybe good point is adding information about content of polyphenols in Medditerranean diet. 

We appreciate this reviewer suggestion and accordingly we revise the introduction section.

Reviewer 2 Report

This manuscript reports findings on the association between urinary and dietary polyphenols and the risk of hip fracture using data of nearly 1500 Italian subjects. The study is well synthesized and written. Few minor comments to improve the manucript are given below:

  1. Introduction: It would be convenient to describe to a greater extent results (and interpretations) on the earlier study on bone health.
  2. Methods: The authors should report whether duplicate assessments of urinary polyphenols were undertaken in order to provide information on the coefficient of variations. This is only reported for vitamin D and PTH determinations. Information on these measurements seem to be not given. Also, it should be clarified whether height and weight was self-reported.
  3. Data analysis: Given that several dietary polyphenols were explored, it may be appropriate to correct for multiple testing. P-values corrected for this issue should be provided.
  4. Results: Information is missing on the correlation between urinary polyphenols and dietary polyphenols in this study. This ifnformation could be important to aid in the interpretation of findings in the dicussion.

Author Response

This manuscript reports findings on the association between urinary and dietary polyphenols and the risk of hip fracture using data of nearly 1500 Italian subjects. The study is well synthesized and written. Few minor comments to improve the manucript are given below:

Introduction: It would be convenient to describe to a greater extent results (and interpretations) on the earlier study on bone health.

We appreciate a lot this reviewer suggestion, and accordingly we reported shortly our interpretations for the earlier study results.

Methods: The authors should report whether duplicate assessments of urinary polyphenols were undertaken in order to provide information on the coefficient of variations. This is only reported for vitamin D and PTH determinations. Information on these measurements seem to be not given.

We agree, with this reviewer, and accordingly we reported the coefficient of variations (lines 109).

Also, it should be clarified whether height and weight was self-reported.

Height and weight were both assessed during the medical visit in every follow-up; accordingly, we corrected the text (lines 162).

Data analysis: Given that several dietary polyphenols were explored, it may be appropriate to correct for multiple testing. P-values corrected for this issue should be provided.

We apology, but in this case, we do not agree, if the suggestion concern table 4, since the correction for multiple comparisons need to be applied when the independent variable has three or more levels. We have assessed differences between subjects reporting a incident hip fracture compared to those subjects who did not experience a fracture during the study times (therefore only 2 levels). Moreover, for every polyphenol (naringenin, kaempferol, luteolin, quercetin, epigallocatechin 3-gallate, and resveratrol) a separate different model was analyzed.

Results: Information is missing on the correlation between urinary polyphenols and dietary polyphenols in this study. This ifnformation could be important to aid in the interpretation of findings in the dicussion.

We agree with this reviewer, and accordingly we have analyzed the correlations between UTP with total dietary assumed Polyphenols across every time of the study, with Linear Mixed models, even if in another paper (referenced in the text) this assumtion was verified.

Reviewers arise a similar observation, we could not know what happened to polyphenols assumption during the 9 years follow-up, as to other confounders.

This is only partially true; we have repeated measures for almost all the variables considered in the manuscript. In our recent paper we have found a decrease in the bone quality related to high UTP levels, but BMD is not enough to predict bone fractures. Therefore, we want to verify if high level of UTP could be a marker of hip fracture increased risk. The problem is similar, for example, to the evaluation of the heart attack risk and smoking status, if the patient stops smoking obviously the heart attack risk decrease; even though smoke habit remained a risk factor.

Unfortunately, we have only basal UTP assessment, whereas for daily P assumption we have a measurement in every follow-up. Therefore, we conducted a new analysis that we reported at the end of the result section, to demonstrate that:

  1. UTP well correlate with P. daily assumption through study times;
  2. P did not change statistically significant through time of the study;
  3. And lastly as we could expect, accordingly to our results, daily dietary-assumption of P, did not differ between patients reporting or not hip fracture.

Reviewer 3 Report

Major comments :

1.The protocol of this prospective   study is complicated, raises several important issues which should be clarified point-by point  :

- Why the samples for UTP were taken only at baseline – at baseline means they were taken after inclusion of subjects into the study and not when the fractures were reported.

              - When pQCT was performed at the beginning of the study ?

              - laboratory tests performed at baseline and at follow-ups ?

2.  In table 3 there is no data on statistical significance for OR which is very strange and makes the interpretation of results impossible ; there is only a statement on page 6, line 236 „the predictive risk of hip-fractures associated with UTP was practically unchanged [OR=2.11; 95%CI (1.15-3.89); p-value =0.01]”.

3. The follow-up was 9 years and during this time at least physical activity, if not some dietary habits, in the study subjects could change with aging. Also several other confounders such as medicines affecting bone metabolism, decreasing vitamin D level with aging due to limited sun exposure in some  individuals under study could influence the results. The authors did not take into account all these confounders when interpreting their results which in my opinion makes the results  not convincing.

4. As mentioned by the authors „higher BMD may not correlate with a reduced incidence of fracture” but higher bone turnover may correlate. Int Osteoporosis Foundation recommends to use bone turnover markers, which reflect metabolism of the whole skeleton,  as predictors of fracture risk in non-treated subjects with osteoporosis. These serum markers are easy to assess and are widely available in the laboratories. Assessment of bone markers was performed in studies dedicated to evaluate the effect of several nutrients on bone health in subjects with osteopenia and osteoporosis. The authors should refer to this important issue in the Discussion.

Minor comments :

There are some typos to be corrected : page 7 line 244 - UTP instead „UPT” and the explanation of abbreviation for „TDP” is missing.

Author Response

The protocol of this prospective   study is complicated, raises several important issues which should be clarified point-by point:

  1. Why the samples for UTP were taken only at baseline – at baseline means they were taken after inclusion of subjects into the study and not when the fractures were reported.
  2. When pQCT was performed at the beginning of the study?
  3. Laboratory tests performed at baseline and at follow-ups?

Laboratory tests, pqct, and 24-h urine sample were collected in every follow-up, moreover the InCHIANTI enrolled 1500 subjects, representative for age and sex of the Italian population distribution.

The UTP assessment is quite expensive, therefore is difficult to repeat UTP evaluation in a total sample of 4000, in the 3 follow-ups.

Lastly, we want to study the hip fracture risk related to UTP levels, independently to classic fracture risk factors as BMD or Muscle mass. Therefore, we want to assess the risk using Logistic Regression Model, and in this approach, we could not consider as independent variable, for example, “the change over time” of BMD; nor is methodologically correct to use the delta between two times of the study (regression to the mean effect).

All reviewers arise a similar observation, we could not know what happened to polyphenols assumption during the 9 years follow-up, as to other confounders.

This is only partially true; we have repeated measures for almost all the variables considered in the manuscript. In our recent paper we have found a decrease in the bone quality related to high UTP levels, but BMD is not enough to predict bone fractures. Therefore, we want to verify if high level of UTP could be a marker of hip fracture increased risk. The problem is similar, for example, to the evaluation of the heart attack risk and smoking status, if the patient stops smoking obviously the heart attack risk decrease; even though smoke habit remained a risk factor.

Unfortunately, we have only basal UTP assessment, whereas for daily P assumption we have a measurement in every follow-up. Therefore, we conducted a new analysis that we reported at the end of the result section, to demonstrate that:

  1. UTP well correlate with P. daily assumption through study times;
  2. P did not change statistically significant through time of the study;
  3. And lastly as we could expect, accordingly to our results, daily dietary-assumption of P, did not differ between patients reporting or not hip fracture.

We did not consider any other confounders variation since in the logistic final model they were not analyzed, because they were not statistically significant different between the two study groups at baseline (David W. Hosmer, Stanley Lemeshow; Applied Logistic Regression. Willey Ed.)

  1. In table 3 there is no data on statistical significance for OR which is very strange and makes the interpretation of results impossible; there is only a statement on page 6, line 236 „the predictive risk of hip-fractures associated with UTP was practically unchanged [OR=2.11; 95%CI (1.15-3.89); p-value =0.01]”.

When OR and 95%CI were used to report Logistic Regression results, p-value was omitted since it is redundant, but if this reviewer believe that p-value is necessary to disentangle the meaning of the analysis, we will report it in the table.

  1. The follow-up was 9 years and during this time at least physical activity, if not some dietary habits, in the study subjects could change with aging. Also several other confounders such as medicines affecting bone metabolism, decreasing vitamin D level with aging due to limited sun exposure in some individuals under study could influence the results. The authors did not take into account all these confounders when interpreting their results which in my opinion makes the results not convincing.

We agree with this reviewer suggestion. As matter of fact, at baseline we analyzed the most important hip fracture risk factors, as physical performance (SPPB), alcohol and smoke habits, muscle strength, BMI (Table 1), pQCT parameters (Table 2), while in ST1 and ST2 the prescribed drugs (at baseline as in the times of the study) and the main diseases reported at baseline as during follow-ups. As suggested by Hosmer and Lemeshow (Applied Logistic Regression. Willey Ed.) we considered in the multivariate Regression Logistic Models only those confounders that at the univariate scored a p-value<0.10.

  1. As mentioned by the authors “higher BMD may not correlate with a reduced incidence of fracture” but higher bone turnover may correlate. Int Osteoporosis Foundation recommends to use bone turnover markers, which reflect metabolism of the whole skeleton,  as predictors of fracture risk in non-treated subjects with osteoporosis. These serum markers are easy to assess and are widely available in the laboratories. Assessment of bone markers was performed in studies dedicated to evaluate the effect of several nutrients on bone health in subjects with osteopenia and osteoporosis. The authors should refer to this important issue in the Discussion.

We appreciate a lot this reviewer’s suggestion. And we accounted for this limitation in the adequate section.

Minor comments :

There are some typos to be corrected : page 7 line 244 - UTP instead „UPT” and the explanation of abbreviation for „TDP” is missing.

We appreciate a lot your effort to ameliorate the manuscript, and accordingly we revised the text as suggested.

Round 2

Reviewer 3 Report

The revised version of the article is enough improved and the answers to reviewers' questions are satisfactory.

Author Response

Common suggestions #1

Both reviewers # 1 e #3 arise a similar observation, we could not know what happened to polyphenols assumption during the 9 years follow-up, as to confounders.

This is only partially true; we have repeated measures for almost all the variables considered in the manuscript. In our recent paper we have found a decrease in the bone quality related to high UTP levels, but BMD is not enough to predict bone fractures. Therefore, we want to verify if high level of UTP could be a marker of hip fracture increased risk. The problem is similar, for example, to the evaluation of the heart attack risk and smoking status, if the patient stops smoking obviously the heart attack risk decrease; even though smoke habit remained a risk factor.

Unfortunately, we have only basal UTP assessment, whereas for daily P assumption we have a measurement in every follow-up. Therefore, we conducted a new analysis that we reported at the end of the result section, to demonstrate that:

  1. UTP well correlate with P. daily assumption through study times;
  2. P did not change statistically significant through time of the study;
  3. And lastly as we could expect, accordingly to our results, daily dietary-assumption of P, did not differ between patients reporting or not hip fracture.

We did not consider any other confounders variation since in the logistic final model they were not analyzed, because they were not statistically significant different between the two study groups at baseline (David W. Hosmer, Stanley Lemeshow; Applied Logistic Regression. Willey Ed.)

Reviewer #1

  1. I suggest adding osteoporosis and osteopenia criteria in the introduction (line 49). 

Accordingly with this reviewer suggestion we insert the definition of osteopenia/osteoporosis t-score, according to WHO.

  1. Why do you use capital letter for name of polyphenols ("among those polyphenols, Naringenin, Kaempferol, Luteolin, Quercetin, Epigallocatechin 3-gallate, and Resveratrol")

We apology, and accordingly to this reviewer suggestion we amended the text. 

  1. You assessed polyphenols intake at baseline and follow-up. However you do not know how many polyphenols subjects intake during 8 years. Maybe, somebody has changed habits after one year or 8 year, which may affects your results. 

Please see common suggestions #1

  1. Introduction is really poor. Maybe good point is adding information about content of polyphenols in Medditerranean diet. 

We appreciate this reviewer suggestion and accordingly we revise the introduction section.

Reviewer #2

This manuscript reports findings on the association between urinary and dietary polyphenols and the risk of hip fracture using data of nearly 1500 Italian subjects. The study is well synthesized and written. Few minor comments to improve the manucript are given below:

  1. Introduction: It would be convenient to describe to a greater extent results (and interpretations) on the earlier study on bone health.

We appreciate a lot this reviewer suggestion, and accordingly we reported shortly our interpretations for the earlier study results.

  1. Methods: The authors should report whether duplicate assessments of urinary polyphenols were undertaken in order to provide information on the coefficient of variations. This is only reported for vitamin D and PTH determinations. Information on these measurements seem to be not given.

We agree, with this reviewer, and accordingly we reported the coefficient of variations (lines 109).

  1. Also, it should be clarified whether height and weight was self-reported.

Height and weight were both assessed during the medical visit; accordingly, we corrected the text (lines 162).

  1. Data analysis: Given that several dietary polyphenols were explored, it may be appropriate to correct for multiple testing. P-values corrected for this issue should be provided.

We apology, but in this case, we do not agree, since the correction for multiple comparisons need to be applied when the independent variable has three or more levels. We have assessed differences between subjects reporting a hip fracture compared to those subjects who did not experience a fracture during the study times (therefore only 2 levels). Moreover, for every polyphenol (naringenin, kaempferol, luteolin, quercetin, epigallocatechin 3-gallate, and resveratrol) a separate different model was analyzed.

  1. Results: Information is missing on the correlation between urinary polyphenols and dietary polyphenols in this study. This ifnformation could be important to aid in the interpretation of findings in the dicussion.

We agree with this reviewer, and accordingly we have analyzed the correlations between UTP with total dietary assumed Polyphenols across every time of the study, with Linear Mixed models (see also common suggestion #1).

Reviewer #3

Major comments :

The protocol of this prospective   study is complicated, raises several important issues which should be clarified point-by point:

  1. Why the samples for UTP were taken only at baseline – at baseline means they were taken after inclusion of subjects into the study and not when the fractures were reported.
  2. When pQCT was performed at the beginning of the study?
  3. Laboratory tests performed at baseline and at follow-ups?

Laboratory tests, pqct, and 24-h urine sample were collected in every follow-up, moreover the InCHIANTI enrolled 1500 subjects, representative for age and sex of the Italian population distribution.

The UTP assessment is quite expensive, therefore is difficult to repeat UTP evaluation in a total sample of 4000, in the 3 follow-ups.

Lastly, we want to study the hip fracture risk related to UTP levels, independently to classic fracture risk factors as BMD or Muscle mass. Therefore, we want to assess the risk using Logistic Regression Model, and in this approach, we could not consider as independent variable, for example, “the change over time” of BMD; nor is methodologically correct to use the delta between two times of the study (regression to the mean effect).

  1. In table 3 there is no data on statistical significance for OR which is very strange and makes the interpretation of results impossible; there is only a statement on page 6, line 236 „the predictive risk of hip-fractures associated with UTP was practically unchanged [OR=2.11; 95%CI (1.15-3.89); p-value =0.01]”.

When OR and 95%CI were used to report Logistic Regression results, p-value was omitted since it is redundant, but if this reviewer believe that p-value is necessary to disentangle the meaning of the analysis, we will report it in the table.

  1. The follow-up was 9 years and during this time at least physical activity, if not some dietary habits, in the study subjects could change with aging. Also several other confounders such as medicines affecting bone metabolism, decreasing vitamin D level with aging due to limited sun exposure in some individuals under study could influence the results. The authors did not take into account all these confounders when interpreting their results which in my opinion makes the results not convincing.

We agree with this reviewer suggestion. As matter of fact, at baseline we analyzed the most important hip fracture risk factors, as physical performance (SPPB), alcohol and smoke habits, muscle strength, BMI (Table 1), pQCT parameters (Table 2), while in ST1 and ST2 the prescribed drugs (at baseline as in the times of the study) and the main diseases reported at baseline as during follow-ups. As suggested by Hosmer and Lemeshow (Applied Logistic Regression. Willey Ed.) we considered in the multivariate Regression Logistic Models only those confounders that at the univariate scored a p-value<0.10.

  1. As mentioned by the authors “higher BMD may not correlate with a reduced incidence of fracture” but higher bone turnover may correlate. Int Osteoporosis Foundation recommends to use bone turnover markers, which reflect metabolism of the whole skeleton,  as predictors of fracture risk in non-treated subjects with osteoporosis. These serum markers are easy to assess and are widely available in the laboratories. Assessment of bone markers was performed in studies dedicated to evaluate the effect of several nutrients on bone health in subjects with osteopenia and osteoporosis. The authors should refer to this important issue in the Discussion.

We appreciate a lot this reviewer’s suggestion. And we accounted for this limitation in the adequate section.

Minor comments :

There are some typos to be corrected : page 7 line 244 - UTP instead „UPT” and the explanation of abbreviation for „TDP” is missing.

We appreciate a lot your effort to ameliorate the manuscript, and accordingly we revised the text as suggested.